# PIEZO1 Promotes the Migration of Endothelial Cells via Enhancing CXCR4 Expression under Simulated Microgravity

**DOI:** 10.3390/ijms25137254

**Published:** 2024-07-01

**Authors:** Yuan Wang, Chengfei Li, Ruonan Wang, Xingcheng Zhao, Yikai Pan, Qian Zhang, Shuhan Li, Jieyi Fan, Yongchun Wang, Xiqing Sun

**Affiliations:** Department of Aerospace Medical Training, School of Aerospace Medicine, Air Force Medical University, Xi’an 710032, China; wangcircle22@163.com (Y.W.); licf920305@163.com (C.L.); xiaopinannan@126.com (R.W.); zhaoxc@fmmu.edu.cn (X.Z.); panyk0820@163.com (Y.P.); zhangqian00525@163.com (Q.Z.); shuhan_li@139.com (S.L.); yiran9911@163.com (J.F.); wangych@fmmu.edu.cn (Y.W.)

**Keywords:** clinorotation, human umbilical vein endothelial cells, PIEZO1, CXCR4, migration

## Abstract

Exposure to microgravity during spaceflight induces the alterations in endothelial cell function associated with post-flight cardiovascular deconditioning. PIEZO1 is a major mechanosensitive ion channel that regulates endothelial cell function. In this study, we used a two-dimensional clinostat to investigate the expression of PIEZO1 and its regulatory mechanism on human umbilical vein endothelial cells (HUVECs) under simulated microgravity. Utilizing quantitative real-time polymerase chain reaction (qRT-PCR) and Western blot analysis, we observed that PIEZO1 expression was significantly increased in response to simulated microgravity. Moreover, we found microgravity promoted endothelial cells migration by increasing expression of PIEZO1. Proteomics analysis highlighted the importance of C-X-C chemokine receptor type 4(CXCR4) as a main target molecule of PIEZO1 in HUVECs. CXCR4 protein level was increased with simulated microgravity and decreased with PIEZO1 knock down. The mechanistic study showed that PIEZO1 enhances CXCR4 expression via Ca^2+^ influx. In addition, CXCR4 could promote endothelial cell migration under simulated microgravity. Taken together, these results suggest that the upregulation of PIEZO1 in response to simulated microgravity regulates endothelial cell migration due to enhancing CXCR4 expression via Ca^2+^ influx.

## 1. Introduction

Microgravity exposure during space expeditions affects human physiological systems, posing significant challenges to interplanetary travel. Microgravity has been found to have a significant effect on the cellular structure and function of endothelial cells, contributing to cardiovascular deconditioning [1]. The vascular endothelial cells are the first barrier to sense mechanical stimulus changes and blood-borne signals and to respond by producing vasoactive signaling molecules such as vascular endothelial growth factor (VEGF), Nitric oxide (NO), and Endothelin-1. Spaceflight changes HUVECs shape and size, remodels the cytoskeleton [2,3]. In turn, endothelial cells dysfunction can lead to cardiovascular deconditioning in space as well as orthostatic intolerance experienced by the majority of astronauts after returning to earth [2,4,5]. Our previous studies found that simulated microgravity promotes cellular migration and autophagy in HUVECs [6,7]. Nevertheless, the connection between mechanosensation and endothelial cell functional alteration under simulated microgravity, as well as the underlying mechanisms, remains largely unknown. 

Endothelial cells sense mechanical stimulus through the cytoskeletal network, and Cadherin-5 and platelet/endothelial cell adhesion molecule-1 [4,8,9,10]. Furthermore, endothelial cells sense and respond to mechanical stimuli primarily via PIEZO1, which is sensitive to mechanical stimulation [11]. PIEZO1 has been recognized for its multifaceted roles in endothelial cells, encompassing the sensing of mechanical stress, regulation of NO release, control of Ca^2+^ influx, promotion of angiogenesis, modulation of vascular tone, etc. [12,13,14,15]. As a non-selective cation channel, PIEZO1 influences endothelial cell behavior by modulating Ca^2+^ influx [12]. PIEZO1 of vascular smooth muscles has been implicated in the mediation of carotid artery remodeling through the upregulation of miR-582-5p following prolonged exposure to simulated microgravity [16]. Although the significance of PIEZO1 in vascular development is evident, its specific contribution to endothelial cell function under simulated microgravity remains to be comprehensively elucidated. 

The migration of endothelial cells is essential not just for the formation of the embryonic vasculature, but also for a spectrum of physiological and pathological processes, including tissue regeneration and wound healing [17]. During angiogenesis, endothelial cells begin to migrate and form vascular sprouts, which then extend to create new vascular lumens [18]. After tissue injury or ischemia, endothelial cells are activated by VEGF and Fibroblast Growth Factor. Activated endothelial cells at the edge of the wound start to migrate over the exposed extracellular matrix towards the site of injury, which is followed by proliferation to cover the denuded area [19]. PIEZO1 has been identified as a pivotal factor in regulating the migration of microglia, keratinocyte, and HUVECs [20,21,22]. Moreover, PIEZO1 promotes the migration of endogenous stem cells in osteoarthritis by mediating the cell-derived factor 1/C-X-C chemokine receptor type 4 (SDF-1/CXCR4) axis [23,24]. Despite extensive research, the precise mechanisms by which PIEZO1 regulates endothelial cell migration under simulated microgravity remain to be fully elucidated.

In this research, we discovered that PIEZO1 is activated via simulated microgravity, which promotes the migration of HUVECs. The analysis of PIEZO1-induced alterations in the HUVECs proteome suggests that CXCR4 may be regulated by PIEZO1, which was verified with Western blot analysis. Furthermore, mechanistic studies revealed that PIEZO1 enhanced CXCR4 expression via Ca^2+^ influx. Surprisingly, CXCR4 inhibition alleviated migration of HUVECs caused by simulated microgravity. In summary, we found that PIEZO1, which was activated with simulated microgravity, promoted the expression of CXCR4 via Ca^2+^ influx, thus enhancing migration capacity of HUVECs under clinorotation. These novel findings provide a better understanding of the molecular and signaling mechanisms regulating HUVECs migration under simulated microgravity.

## 2. Results

### 2.1. Simulated Microgravity Increases the Expression of PIEZO1 in HUVECs

In this study, we conducted an in vitro experiment using HUVECs cultured in clinostat after clinorotation for 6 h, 12 h, 24 h, and 48 h. To identify the effect of simulated microgravity on PIEZO1, qRT-PCR and Western blot were used to measure expression levels of PIEZO1. As observed in Figure 1A, the mRNA level was increased significantly after clinorotation for 12 h, 24 h, and 48 h. As shown in Figure 1B, clinorotation for 12 h, 24 h, and 48 h also increased PIEZO1 expression at the protein level in a time-dependent manner. Compared with the control group, PIEZO1 expression level at the protein and mRNA levels increased gradually with exposure time. Consequently, we chose 48 h as the time point for further analysis due to qRT-PCR and Western blot consistently showing the highest expression level of PIEZO1 at clinorotation for 48 h. Moreover, immunofluorescence demonstrated that the expression of PIEZO1 increased after simulated microgravity (MG) for 48 h, which was consistent with qRT-PCR and Western blot analysis (Figure 1C). The data indicated that PIEZO1 was upregulated in HUVECs under simulated microgravity.

### 2.2. Simulated Microgravity Promotes Endothelial Cell Migration through PIEZO1

Endothelial cell migration is a crucial process throughout life, including embryogenesis, vessel repair, and angiogenesis [25,26,27]. Our previous study confirmed that clinorotation for 48 h increases HUVEC migration [7]. To investigate whether simulated microgravity promotes endothelial cell migration through PIEZO1, we used PIEZO1 siRNA to knock down PIEZO1 under the clinorotation condition. The interference efficiency of siRNA3 was higher than that of the other two PIEZO1 siRNAs (siRNA1 and siRNA2), which were used in subsequent experiments (Appendix A). A Transwell assay of HUVECs showed that the number of migrated cells in the microgravity group with si-NC treatment (MG+si-NC) was approximately 1.6-fold of the Con+si-NC group. Nevertheless, the si-PIEZO1 treatment reduced the number of migrated cells (Figure 2A). Similarly, compared to the Con+si-NC group, the rate of wound healing in the MG+si-NC group was higher, and si-PIEZO1 treatment under simulated microgravity alleviated HUVECs’ wound-healing ability (Figure 2B). These results suggested that simulated microgravity promotes endothelial cell migration through PIEZO1.

### 2.3. PIEZO1 Knock-Down in HUVECs after Exposure to Simulated Microgravity Results in Changes in Several Biological Processes

To better understand the molecular mechanisms by which PIEZO1 regulates endothelial cell migration under simulated microgravity, we performed proteomic profile analysis of HUVECs using the LC-MS/MS method. For this analysis, we collected HUVEC samples from the Con+si-NC group, MG+si-NC group, and MG+si-PIEZO1 group to make a total protein extraction and proteomics analysis. The principal component analysis revealed a high level of repeatability among the samples in each group (Figure 3A). A total of 52,087 peptides were identified with high confidence (false discovery rate < 1%), of which 49,523 peptides were identified as unique peptides. We identified 6408 proteins for subsequent analysis (Appendix A). A stringent criterion was applied for quantitative analysis, and we set the threshold for significant upregulation to be higher than 1.5 and the threshold for significant downregulation to be less than 1/1.5 (*p* < 0.05). LC-MS/MS analysis identified a total of 197 DEPs between the MG+si-NC and Con+si-NC groups, with 121 proteins downregulated and 76 upregulated. Additionally, a comparison between the MG+si-PIEZO1 and MG+si-NC groups revealed 212 DEPs, consisting of 85 downregulated and 127 upregulated proteins. (Figure 3B,C). 

To elucidate the mechanisms reflected by PIEZO1-induced proteome changes under simulated microgravity, we used the Mfuzz method for cluster analysis of protein abundance transformations in different groups. A total of 6 different expression patterns were clustered and their expression levels were visualized using a heat map (Figure 3D). The GO terms of each cluster were plotted according to the fold enrich value from the largest to smallest and the *p* value (*p* < 0.05). GO analysis revealed that the cluster5 group exhibited the highest level of enrichment and was enriched in biological processes (BP), such as “cell activation”, “cellular response to chemical stimulus”, and “cell activation involved in immune response”; in molecular functions (MF), such as “cation binding”, “metal ion binding” and “protein-containing complex binding”; in cellular components (CC), such as “cell periphery”, “plasma membrane”, and “endoplasmic reticulum membrane”; etc. (Figure 3E). A Venn diagram compared the significant DEPs of MG+si-NC vs. Con+si-NC and MG+si-PIEZO1 vs. MG+si-NC, respectively. Among these DEPs, we found 10 and 39 overlapping significantly enriched proteins between MG+si-NC vs. CON+si-NC up and MG+si-PIEZO1 vs. MG+si-NC down, MG+si-NC vs. CON+si-NC down and MG+si-PIEZO1 vs. MG+si-NC up, respectively (Figure 3F).

Through the above analysis, we noticed that CXCR4, identified in cluster5 from cluster analysis, which plays critical roles in regulating endothelial cell function, was associated with pathways identified in GO analysis (Figure 3E). In addition, we found CXCR4 was upregulated in MG+si-NC vs. CON+si-NC, whereas it was downregulated in MG+si-PIEZO1 vs. MG+si-NC (Appendix A). It is well known that activation of the SDF-1α/CXCR4 axis is widely recognized to play a crucial role in migration of endothelial cells [28]. We thus speculated that CXCR4 may play a predominant role in PIEZO1-promoted endothelial cell migration under simulated microgravity.

### 2.4. PIEZO1 Enhances the Expression of CXCR4 through Ca^2+^ Influx

To confirm the proteomics findings indicating that CXCR4 expression is upregulated through microgravity and downregulated by si-PIEZO1 treatment, we performed Western blot analysis. Consistent with the proteomics analysis, the expression of CXCR4 increased after simulated microgravity, but decreased after knocking down PIEZO1 (Figure 4A). Ca^2+^ is an important intracellular second messenger involved in the regulation of many cellular events [29]. Therefore, the capacity of PIEZO1 to enhance CXCR4 expression may be attributed to its ion channel function, which enables the Ca^2+^ influx upon mechanical stimulation. To illustrate the above point, we first used Fluo-4AM staining to detect the intracellular Ca^2+^ of HUVECs. Exposure to simulated microgravity led to a significant increase in the Ca^2+^ levels of HUVECs, which was dramatically attenuated by PIEZO1 knockdown treatment (Figure 4B). BAPTA-AM, a permeable calcium chelator, was used to perturb the calcium-dependent processes. As expected, pretreatment with BAPTA-AM significantly reduced the elevation of CXCR4 protein levels that were induced through simulated microgravity (Figure 4C). These results indicated that PIEZO1 significantly enhanced the expression of CXCR4 via Ca^2+^ influx under simulated microgravity.

### 2.5. CXCR4 Promotes the Migration of HUVECs under Simulated Microgravity

To ascertain whether CXCR4 facilitates cell migration under simulated microgravity, we conducted both Transwell and wound healing assays. We used AMD3100, an inhibitor of CXCR4, to treat HUVECs, which were exposed to simulated microgravity for 48 h. Compared to the control group, the number of migrated cells was increased in the MG group, which was decreased by AMD3100 (Figure 5A). In concordance with the Transwell assay results, wound-healing assays showed that AMD3100 effectively reduced CXCR4-driven cell migration under simulated microgravity (Figure 5B). These data confirmed that CXCR4 promoted the migration of HUVECs under simulated microgravity.

## 3. Discussion

In this study, we confirmed the promotion effect of PIEZO1 on the migration of endothelial cells under simulated microgravity, identified PIEZO1-induced proteome changes in endothelial cells, and proved that CXCR4 served as a downstream target of PIEZO1 that mediated PIEZO1-enhanced endothelial cell migration. In addition, for the first time, we showed that upregulation of PIEZO1 induced by simulated microgravity increased CXCR4 protein level through Ca^2+^ influx, which subsequently promoted migration of endothelial cells as shown in Figure 6.

The microgravity in space causes a change in mechanical stress, resulting in some changes including a decrease in stroke volume, increased heart rate, a decrease in left ventricular volume, increased vascular intima-media thickness, and carotid artery plaque formation [30,31,32]. Current research indicates that the PIEZO1/2 cationic mechanosensitive ion (MS) channels have a key role in multiple aspects of cardiovascular development and physiology. PIEZO1 can sense various mechanical stresses, including gravity, shear stress, and membrane stretch, mainly in non-sensory tissues and non-neuronal cells [33]. A majority of PIEZO2 is found in the sensory system, such as the dorsal root ganglion sensory neurons and Merkel cells, and it regulates mechanical nociception [34,35]. Previous research demonstrated that PIEZO1 expression in endothelial cells was regulated by the NF-kappa B signaling pathway, where TNF-α activated NF-kappa B through p65 signaling, subsequently upregulating PIEZO1 expression [36]. In our study, we have observed that simulated microgravity conditions lead to an upregulation of PIEZO1 at both the mRNA and protein levels, as confirmed with qPCR and Western blot analysis. Despite these observations, the underlying mechanisms by which simulated microgravity induces the upregulation of PIEZO1 are not yet fully understood and warrant further investigation. PIEZO1 also acts as an important mechanosensor required for early vascular development and morphogenesis of valve mediates, as well as endothelial flow-dependent vasodilatation and atheroprotection [15,37]. However, the specific target function of endothelial cell PIEZO1 remains unclear in microgravity. In our study, we first demonstrated that PIEZO1 was highly expressed in HUVECs after exposure to simulated microgravity.

Due to mechanical stress changes, endothelial cell dysfunction is a common phenomenon under microgravity. Our previous research revealed that simulated microgravity induces a variety of morphological and functional changes in HUVECs, including cytoskeletal remodeling, migration, autophagy, and apoptosis [6,7,38]. A significant finding in this study was that simulated microgravity could increase PIEZO1 expression in HUVECs, which was confirmed with qRT-PCR and Western blot analysis. In addition to the PIEZO1 activation discussed above, we also found PIEZO1 could promote the migration of HUVECs under simulated microgravity. However, some studies reported that simulated microgravity treatment suppressed PIEZO1 expression in osteoblast and bone marrow MSCs, which impaired osteoblast function and led to insufficient bone formation [39,40]. Interestingly, it has been documented that SMC-PIEZO1-KO mice were resistant to simulated microgravity-induced carotid stiffness and thickening [16]. Our results indicated that PIEZO1 plays an important role in regulating endothelial cell migration under simulated microgravity.

As a non-selective cation channel, PIEZO1 regulates endothelial cell functions that are heavily dependent on intracellular calcium ions, which subsequently trigger signaling cascades [41]. Previous studies reported that the PIEZO1-mediated calcium-activated calpain/VE-cadherin pathway promotes disruption of the pulmonary endothelial barrier [9]. Several studies have shown that simulated microgravity can cause cellular Ca^2+^ oscillations [42,43]. Both simulated microgravity and hypergravity increased Ca^2+^ in cardiomyocytes, which regulated myocardial remodeling through the CaMKII/HDAC4 signaling pathway. Our work here demonstrated that intracellular Ca^2+^ levels of HUVECs were increased under simulated microgravity, while PIEZO1 knockdown reduced the Ca^2+^ levels in the cells. Endothelial cell migration was synchronized with the changes in Ca^2+^ concentration. These results suggested that calcium signaling is a major regulating mechanism for the homeostasis of mechano-induced endothelial cells.

Despite extensive research aimed at elucidating the downstream effects of PIEZO1 across various cell types, including erythrocytes [44] and glioma cells [45], at both the genomic and transcriptomic levels, the proteomic changes induced via PIEZO1 knockdown, particularly in endothelial cells under simulated microgravity, have not yet been explored. Investigating these changes is crucial for understanding the mechanisms by which PIEZO1 regulates endothelial cell function following exposure to microgravity conditions. Through proteomics analysis, we found that, under simulated microgravity conditions, PIEZO1 regulates several important signaling pathways of endothelial cell function, including cell activation, cell death, regulation of response to wounding and cell periphery, etc. We identified 197 DEPs between MG+si-NC groupand CON+si-NC, group while 212 proteins showed significant changes between MG+siPIEZO1 group and MG+si-NC group. Moreover, we found there were a total of 51 DEPs between the MG+si-NC vs. CON+si-NC group and MG+si-PIEZO1 vs. MG+si-NC group. Among these DEPs, we noticed CXCR4, of which the fold change is the most obvious.

CXCR4 is a member of the CXCR chemokine receptor family and can be activated by SDF-1α, which is abundantly expressed on endothelial cells [46]. CXCR4 is regulated by PIEZO1 in endogenous stem cells [23,24]. CXCR4 plays a pivotal role in directing the migration of endothelial cells and preserving endothelial integrity [47]. The activation of CXCR4 activates a number of downstream signaling pathways, such as the MEK/ERK pathway, the PI3K/Akt/mTOR pathway, NF-B signaling, etc. These pathways contribute to vasculopathy, local inflammation, chemotaxis, and proliferation [46]. There is evidence that targeting SDF-1/CXCR4 is an effective treatment for certain types of tumors, such as glioblastomas and melanoma [48,49]. Blocking CXCR4 can affect tumor cell metastasis by causing receptor internalization and the dispersion of adhesion proteins, ultimately leading to a decrease in intercellular adhesion. It has been established that the upstream molecules of CXCR4 have not been determined, despite a large number of studies. In static magnetic fields, PIEZO1 activates SDF-1 and CXCR4 to enhance Bone marrow mesenchymal stem cells (BMSCs) migration abilities [23]. Additionally, PIEZO1 is known to modulate the SDF-1/CXCR4 axis to facilitate the migration of BMSCS during osteoarthritis [24]. Similarly, our study demonstrated that PIEZO1 regulates endothelial cell migration by activating CXCR4 in response to simulated microgravity. Based on proteomics results, simulated microgravity could promote the expression of CXCR4, while knocking down PIEZO1 inhibited its expression. This was confirmed with Western blot analysis. There was a surprising finding that PIEZO1 enhanced CXCR4 expression by increasing intracellular Ca^2+^ levels. These results thus reveal a novel mechanism by which PIEZO1 is a mediator for CXCR4 enhancing endothelial cell migration and provide new insights into vascular dysfunction under simulated microgravity.

Above all, this study verifies that simulated microgravity activates PIEZO1, which in turn enhances the Ca^2+^ influx of HUVECs. The increased Ca^2+^ in HUVECs promotes expression of CXCR4, thus enhancing cell migration capacity. This study has novel implications for the mechanisms of cardiovascular dysfunction under simulated microgravity.

## 4. Materials and Methods

### 4.1. Cell Culture and Drug Treatment

The HUVECs were sourced from American Type Culture Collection (ATCC, Manassas, VA, USA) and cultured in high-glucose Dulbecco’s modified Eagle’s medium (DMEM, Hyclone, Logan, UT, USA), supplemented with 10% fetal bovine serum (FBS, Hyclone, Logan, UT, USA). The cells were incubated at 37 °C in an environment with 5% CO_2_ and 95% relative humidity. The growth medium was exchanged every 48 h to ensure optimal cell health. Upon achieving approximately 80% confluence, the cells were treated with trypsin to detach them from the culture surface, thus enabling the subsequent experimental procedures. Intracellular calcium levels were measured using the fluorescent indicator Fluo4-acetoxymethyl ester (Fluo4-AM, Beyotime, Shanghai, China) at a final concentration of 5 μM. In order to eliminate the influence of calcium in HUVECs, we incubated cells with a membrane-permeable calcium chelator, BAPTA-AM (MCE, Monmouth Junction, NJ, USA), at a concentration of 10 μM, which was dissolved in dimethyl sulfoxide (DMSO). Additionally, the CXCR4 antagonist AMD3100 (MCE, Monmouth Junction, NJ, USA), dissolved in double-distilled water (ddH_2_O), was utilized to inhibit the CXCR4 receptor at a concentration of 50 μM.

### 4.2. Clinorotation to Simulate Microgravity

2D clinorotation (developed by the China Astronaut Research and Training Centre, Beijing, China) is widely used to simulate a microgravity environment for cells on the ground, and the effectiveness of simulating microgravity has been demonstrated [1,2]. The cells were seeded at a density of 1 × 10^5^ cells on 2.55 cm × 2.15 cm coverslips within six-well culture plates at 37 °C in a humidified incubator with an atmosphere containing 5% CO_2_. Once the cells had firmly adhered to the substrate, the coverslips were carefully transferred to a dedicated chamber pre-filled with culture medium. Subsequently, the chamber was placed into a clinostat system for incubation, where it underwent continuous rotation at a rate of 30 revolutions per minute (rpm) over a range of durations. This rotation rate effectively simulated a gravitational force of approximately 10^−3^ g on the HUVECs, replicating the microgravity conditions. The cells incubated within the clinostat system constituted the simulated microgravity group (MG). The control group (Con) was cultured under identical conditions within the same chamber without clinorotation. All experimental procedures were uniformly conducted at a temperature of 37 °C.

### 4.3. qRT-PCR Analysis

Total RNA was isolated from HUVECs using the TRIzol Reagent (Invitrogen, Carlsbad, CA, USA) according to the manufacturer’s protocol. Subsequently, 0.5 ug of total RNA was reverse-transcribed to cDNA using the PrimeScriptRT reagent Kit (Takara, Tokyo, Japan). The subsequent real-time PCR detection was conducted using SYBR^®^ Premix Ex TaqTM II (YISHEN, Shanghai, China). The amplification and detection of the cDNA were carried out on a Roche LightCycler 480 system (Roche, Manheim, Germany). The PCR cycling parameters were as follows: an initial denaturation at 95 °C for 30 s, followed by 40 cycles of denaturation at 95 °C for 5 s and extension at 60 °C for 40 s. GAPDH was used as an endogenous reference gene to normalize the data. The relative quantification of mRNA expression levels between samples was determined using the relative Ct (2^−ΔΔCt^) method and were expressed as a fold change compared with the endogenous control GAPDH. The sequences of the primers are presented as follows:

5′-CCTGGAGAAGACTGACGGCTAC-3′/5′-ATGCTCCTTGGATGGTGAGTCC-3′ (human PIEZO1) and 5′-GTCTCCTCTGACTTCAACAGCG-3′/5′-ACCACCCTGT TGCTGTAGCCAA-3′ (human GAPDH).

### 4.4. Western Blot Analysis

Total protein was extracted from cells using RIPA buffer (NCM Biotech, Suzhou, China) supplemented with 1 mM phosphatase inhibitor cocktail on ice. Protein concentration was measured with the BCA assay (Beyotime, Shanghai, China). The total protein samples were boiled with loading buffer (Bio-Rad, CA, USA) for 10 min. Following this, the samples were subjected to sodium dodecyl sulfate-polyacrylamide gel electrophoresis (SDS-PAGE) separation and then transferred onto a 0.45 μm polyvinylidene fluoride (PVDF) membrane. The membranes were blocked with 5% non-fat milk in a Tris-buffered saline-Tween 20 (TBST) solution at room temperature for 2 h. After blocking, membranes were incubated with the following primary antibodies: anti-PIEZO1 (1:1000, Abcam, Cambridge, UK, ab259949) and CXCR4 (1:1000, Abcam, Cambridge, UK, ab124824) and anti-GAPDH (1:2000, Zhuangzhi Biology, Xian, China, NC021), overnight at 4 °C to allow for antibody binding. The membranes were incubated with horseradish peroxidase (HRP)-conjugated secondary antibody for 1 h at room temperature. Protein bands were detected using an enhanced chemiluminescence (Millipore, Danvers, MA, USA). The band intensity was quantified with ImageJ (https://imagej.nih.gov/ij/index.html, accessed on 25 June 2024) and normalized to GAPDH as a loading control.

### 4.5. Immunofluorescence Staining

Cells were first fixed using a 4% solution of paraformaldehyde for 10 min and then permeabilized with 0.5% Triton X-100 in phosphate-buffered saline (PBS) for 30 min at room temperature. Following this, non-specific binding was minimized by blocking with 10% normal goat serum. The cells were then incubated with PIEZO1 antibody (1:200; Proteinteach, Wuhan, China) overnight at 4 °C. On the subsequent day, the cells were treated with a secondary antibody conjugated with Alexa Fluor 488 (1:1000; Beyotime, Shanghai, China) for 1 h at room temperature. Nuclear staining was conducted in the absence of light using DAPI. The fluorescence from the stained cells were subsequently visualized via CaseViewer 2.4 (CaseViewer 2.4; 3DHISTECH, Budapest, Hungary).

### 4.6. Transfection of Small Interfering RNA

Cells for RNA interference were transfected with PIEZO1 siRNA (si-PIEZO1) or negative control siRNA (si-NC) at 70% confluence using Lipofectamine 2000 (Invitrogen, USA) according to the manufacturer’s protocol. Sequences of the siRNA probes were as follows: si-NC, 5′UUCUCCGAACGUGUCACGUTT-3′; si-PIEZO1, 5′-CACCGGCATCTACG TCAAATA-3′. The transfection efficiency was detected via qPCR and Western blot after transfection for 48 h. Piezo1 siRNA was cultured with HUVECs before exposure to simulated microgravity.

### 4.7. Cytosolic Ca^2+^ Measurements

Ca^2+^ imaging experiments were conducted using the Fluo4-acetoxymethyl ester (Fluo4-AM, Beyotime, China). Cells were incubated with 5 μM Fluo-4 AM in Hanks’ balanced salt solution (HBSS; 0.137 M NaCl, 5.4 mM KCl, 0.25 mM Na_2_HPO_4_, 0.44 mM KH_2_PO_4_, 4.2 mM NaHCO_3_, 5.56 mM glucose, and 10 mM HEPES, pH 7.4) without Ca^2+^ and Mg^2+^ for 30 min at room temperature. Following a 30 min incubation at room temperature with Fluo4-AM, the cells were washed twice with HBSS before fluorescence measurements. Fluorescence measurements from single cells were captured with the LSM 800 confocal fluorescence microscope (Zeiss, Oberkochen, Germany). Fluo-4 was excited at 488 nm, and the emission was collected through a 505–550 nm barrier filter. The resulting images were processed and analyzed using ImageJ (https://imagej.nih.gov/ij/index.html, accessed on 25 June 2024). The data presented were normalized to the baseline fluorescence for comparison.

### 4.8. Transwell Assays

The Transwell assay was conducted using Transwell Permeable Supports (Corning Inc., Corning, NY, USA). Transwell chambers were placed into the 24-well culture plate. After trypsinization, cells were collected and a suspension was prepared with 3 × 10^4^ cells in 100 μL of medium containing 0.25% FBS, which was seeded into the upper chamber. And the lower chamber was filled with 600 μL of medium containing 10% FBS. After a 12 h incubation period, 4% paraformaldehyde was used to fix the migrated cells for 10 min. Subsequently, the cells were stained with a 0.4% crystal violet solution. Next, the cells that remained in the upper surface of polycarbonate membrane were removed with a cotton swab. The migrated cells were captured using a microscope (Nikon, Tokyo, Japan).

### 4.9. Wound-Healing Assay

After simulated microgravity for 48 h, each coverslip was scratched with a sterilized 200 mL pipette tip. HUVECs were then cultured in serum-free medium instead of complete media. An inverted microscope was used to measure wound widths at 0 h and 12 h. The mean width was precisely determined with the ImageJ software (https://imagej.nih.gov/ij/index.html, accessed on 25 June 2024). Based on these measurements, the rates of scratch wound healing were subsequently calculated.

### 4.10. Mass Spectrometric Analysis

#### 4.10.1. Sample Preparation

The protein lysates were collected in three biological replicates of HUVECs from three groups including the Con+si-NC group, MG+si-NC group, and MG+si-PIEZO1 group. For each sample group, lysate was added(8 M urea, 1% protease inhibitor, and 1% phosphatase inhibitor), followed by sonication for three minutes on ice. Afterward, the mixture was centrifuged at 4 °C at 12,000× *g* for 10 min. The cellular debris was removed, and the supernatant was transferred to a new centrifuge tube. The protein concentration was then determined using a BCA assay kit. Equal amounts of protein from each sample were subjected to enzymatic digestion. The volume of the samples was adjusted to uniformity using lysis buffer, and then a final concentration of 20% TCA was slowly added with vortex mixing. The mixture was allowed to precipitate at 4 °C for 2 h. The samples were then centrifuged at 4500× *g* for 5 min, the supernatant was discarded, and the precipitated protein was washed 2–3 times with pre-chilled acetone. After one minute of drying it was resuspended in TEAB at a final concentration of 200 mM. For the digestion overnight, trypsin was added at a ratio of 1:50 (protease: protein, m/m). Dithiothreitol (DTT) was added to achieve a final concentration of 5 mM, and the mixture was reduced at 56 °C for 30 min. Subsequently, Iodoacetamide (IAA) was introduced to reach a final concentration of 11 mM, and the sample was incubated in the dark at room temperature for 15 min. Finally, the peptides were desalted using a Strata X solid-phase extraction column.

#### 4.10.2. Liquid Chromatography and Tandem Mass Spectrometry (LC–MS/MS)

Peptides were dissolved in solvent A and injected directly onto a custom-made reversed-phase analytical column (25-cm length, 100 μm i.d.). The mobile phase included solvent A (0.1% formic acid and 2% acetonitrile/in water) and solvent B (0.1% formic acid and 90% acetonitrile/in water). A gradient (0–22.5 min, 6–22%; 22.5–26.5 min, 22–34%B; 26.5–28.5 min, 34–80%B; and 28.5–30 min, 80%B) was used to separate Peptides. This process was at a steady flow rate of 700 nL/min on an EASY-nLC 1200 UPLC system (ThermoFisher Scientific, Waltham, MA, USA). Using a nano-electrospray ion source, the separated peptides were analyzed on an Orbitrap Exploris 480. FAIMS compensate voltage (CV) was set to 45 V and Electrospray voltage was set at 2300 V. The Orbitrap detector was used to analyze precursors and fragments. Using a 350–1400 *m*/*z* scan range, the full MS scan resolution was set to 60,000. With a resolution of 15,000, the first mass of the MS/MS scan was fixed at 120.0 *m*/*z*. In this experiment, the normalized collision energy (NCE) of the HCD fragments was 27%. Using an automatic gain control (AGC) target of 10^6^, the injection time was limited to 22 ms.

#### 4.10.3. Data Processing

To build the Spectral Library, data from the Data Dependent Acquisition (DDA) were processed using Spectronaut (v.17.0) software combined with the Pulsar search engine. The tandem mass spectra were compared against Homo_sapiens_9606_SP_20230103 fasta (20,389 entries) concatenated with reverse decoy data. The max missing cleavages was set as 2. PSM, peptide, and protein false discovery rates (FDR) were adjusted to 1%. In Spectronaut (v.17.0) software, the library of corresponding spectra is imported and the retention time is predicted using nonlinear correction and also compared with DIA data. To predict the retention time with nonlinear correction, the corresponding spectral library was imported into Spectronaut (v.17.0) software. This library was then searched against with Data Independent Acquisition (DIA) data.

#### 4.10.4. Bioinformatics Analysis

The clustering analysis was carried out using the “Mfuzz” R package. Gene ontology (GO) and Pathway analysis was performed using the eggnog-mapper software (http://eggnog-mapper.embl.de) to extract GO IDs from the identified proteins in the EggNOG database. The significance of functional enrichment of differentially expressed proteins (DEPs) was analyzed via Fisher’s exact test. Fold enrichment values > 1.5 and *p* value < 0.05 were considered significant.

### 4.11. Statistical Analysis

Data analysis was conducted using GraphPad Prism software(8.3.0.). The results are presented as the mean ± *SEM* of three independent experiments. For multiple comparisons of more than two normally distributed groups, one way ANOVA with Dunnett post-test was used. *p* < 0.05 was considered statistically significant.

## 5. Conclusions

Most importantly, our research has revealed that simulated microgravity induces the activation of PIEZO1 in HUVECs. This activation subsequently leads to an increased influx of Ca^2+^ into the cells. The elevated intracellular calcium levels facilitate the upregulation of the chemokine receptor CXCR4, which in turn significantly enhances the migratory capacity of the cells. Our findings provide novel insights into the mechanisms underlying cardiovascular dysfunction in conditions that mimic microgravity, offering potential avenues for future research and therapeutic intervention.

## Figures and Tables

**Figure 1 ijms-25-07254-f001:**
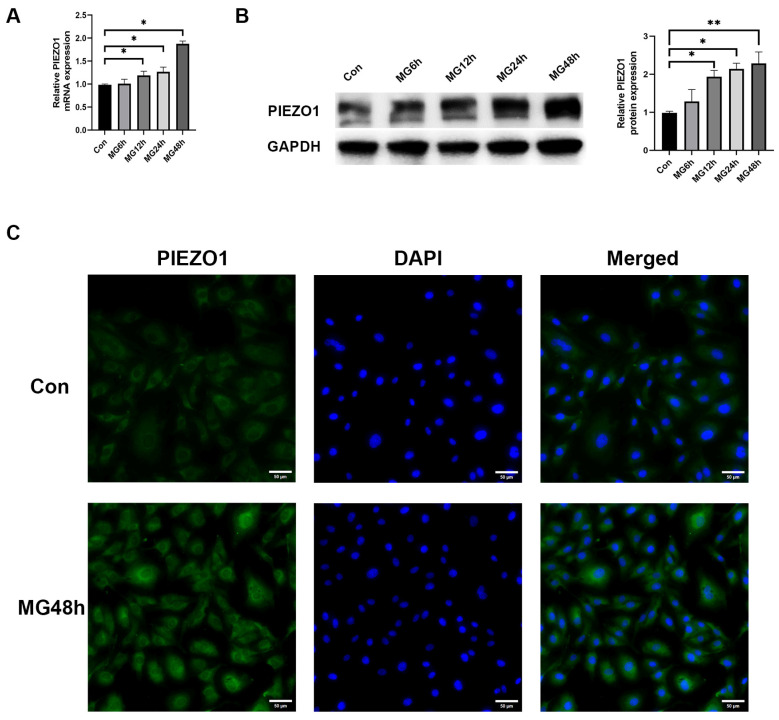
PIEZO1 expression in HUVECs is elevated in simulated microgravity conditions. (**A**) qRT-PCR analysis of PIEZO1 in HUVECs after clinorotation for 6, 12, 24, and 48 h. (**B**) Western blot analysis for the expression of PIEZO1 in HUVECs after clinorotation for 6, 12, 24, and 48 h. (**C**) The immunofluorescence staining of PIEZO1 (greenimmunofluorescence) and DAPI (blue immunofluorescence). (The data represent the mean ± *SEM*, *n* = 3; * *p* < 0.05, ** *p* < 0.01).

**Figure 2 ijms-25-07254-f002:**
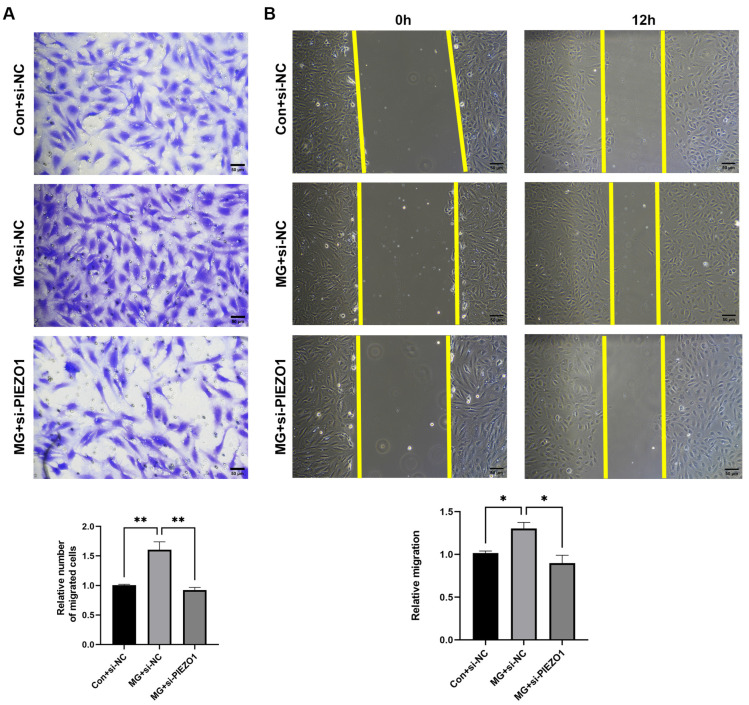
Simulated microgravity enhances endothelial cell migration via the PIEZO1 channel. (**A**) Transwell migration assay. Cells were treated with or without si-Piezo1 while exposed to microgravity for 48 h, and then a Transwell migration assay was performed, and the migrated cells were fixed and stained. Representative photos of migrated HUVECs were observed using the microscope. (**B**) Wound-healing assay. Cells were exposed to simulated microgravity for 48 h with or without si-PIEZO1. Then, scratches were made and cultured for 12 h. Microphotographs of the scratches were obtained as soon as the scratches were made and after 12 h. (The data represent the mean ± *SEM*, *n* = 3; * *p* < 0.05, ** *p* < 0.01).

**Figure 3 ijms-25-07254-f003:**
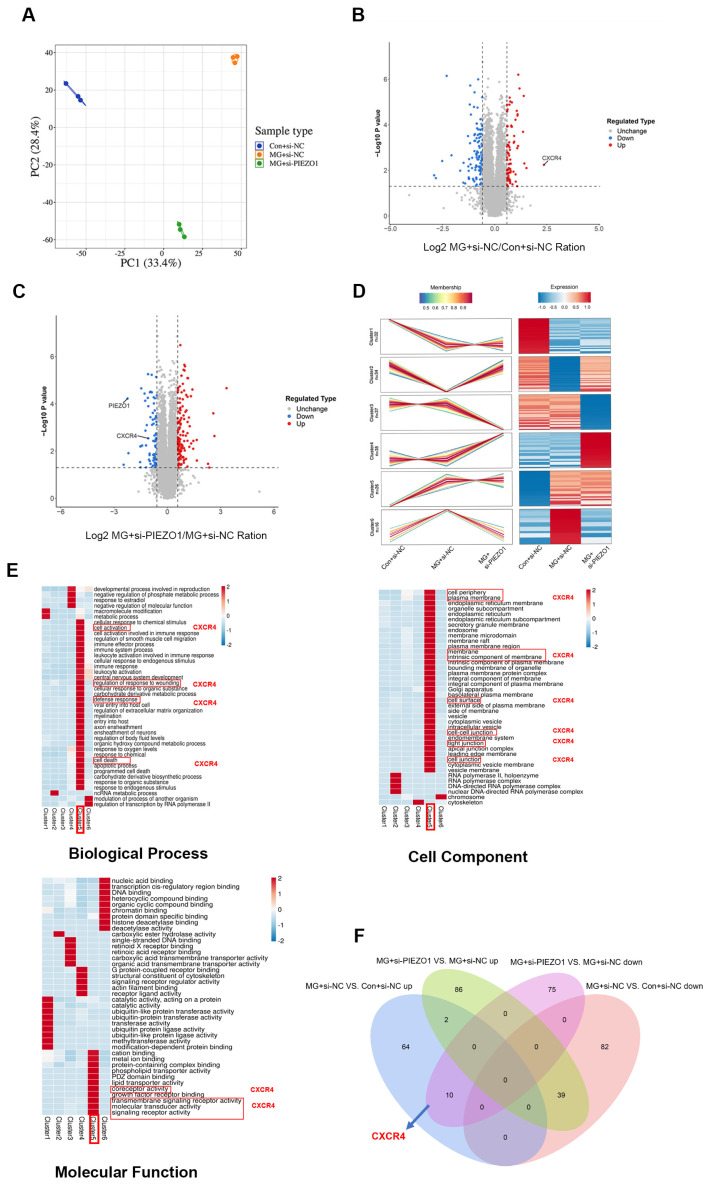
The knockdown of PIEZO1 in HUVECs leads to modifications in various biological processes under simulated microgravity. (**A**) Principal component analysis of the relative quantitative values of all samples. HUVECs were transfected with si-NC and si-PIEZO1. HUVECs with si-NC were exposed to simulated microgravity for 48 h (MG+si-NC) and under stationary condition for 48 h (Con+si-NC). HUVECs with si-PIEZO1 were exposed to simulated microgravity for 48 h (MG+si-PIEZO1). Proteins isolated from HUVECs were subjected to LC-MS/MS analysis. (**B**,**C**) Volcanic maps of differential proteins, where red dots indicate significant upregulation, blue dots indicate significant downregulation, and gray indicates no significant differences. (**D**) Mfuzz method for cluster analysis of protein abundance transformations in different groups. (**E**) The GO terms of Mfuzz method cluster analysis. (**F**) A Venn diagram compared the significant DEPs of MG+si-NC vs. CON+si-NC and MG+si-PIEZO1 vs. MG+si-NC. *(n* = 3).

**Figure 4 ijms-25-07254-f004:**
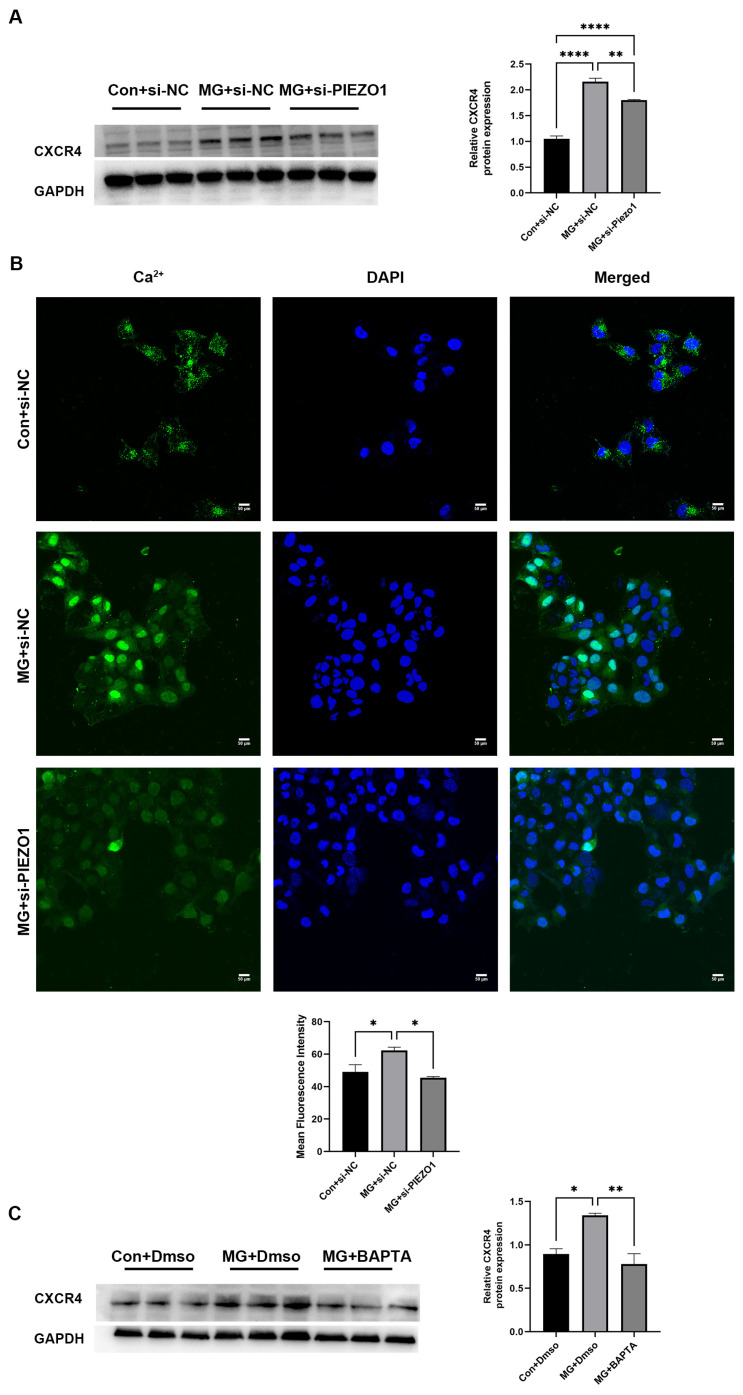
PIEZO1 induces the upregulation of CXCR4 expression via Ca^2+^ influx. (**A**) Western blot analysis for the expression of CXCR4 in HUVECs with or without si-PIEZO1 while exposed to microgravity for 48 h. (**B**) Ca^2+^ influx was detected with confocal microscopy and the quantitation of immunofluorescence. The immunofluorescence staining of Ca^2+^(green immunofluorescence) and DAPI (blue immunofluorescence). (**C**) Western blot analysis for the expression of CXCR4 in HUVECs with or without BAPTA while exposed to microgravity for 48 h. (The data represent the mean ± *SEM*, *n* = 3; * *p* < 0.05, ** *p* < 0.01, **** *p* < 0.0001).

**Figure 5 ijms-25-07254-f005:**
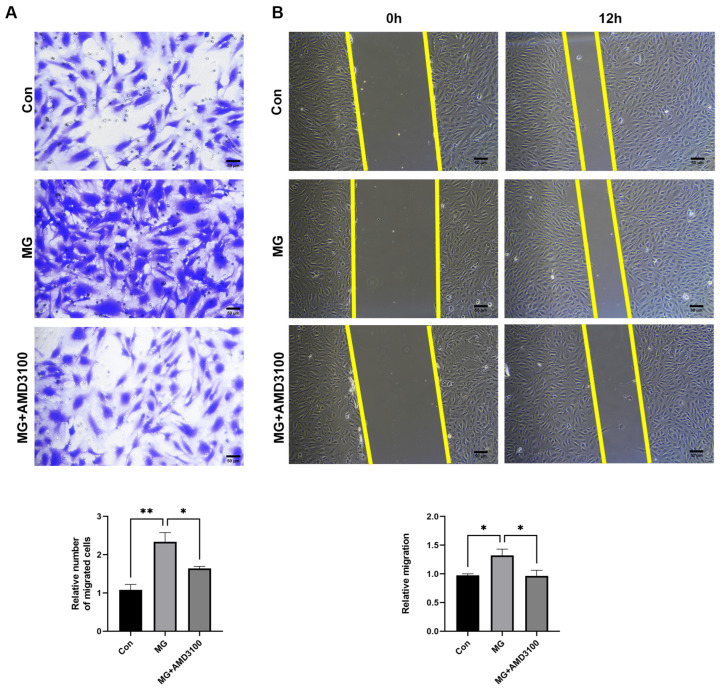
The migration of HUVECs is enhanced by CXCR4 under simulated microgravity. (**A**) Transwell migration assay. Cells were treated with or without AMD3100 while exposed to microgravity for 48 h, and then a Transwell migration assay was performed, and the migrated cells were fixed and stained. Representative photos of migrated HUVECs were observed using the microscope. (**B**) Wound-healing assay. Cells were exposed to simulated microgravity for 48 h with or without AMD3100. Then, the scratches were made and cultured for 12 h. Microphotographs of the scratches were obtained as soon as the scratches were made and after 12 h. (The data represent the mean ± *SEM*, *n* = 3; * *p* < 0.05, ** *p* < 0.01).

**Figure 6 ijms-25-07254-f006:**
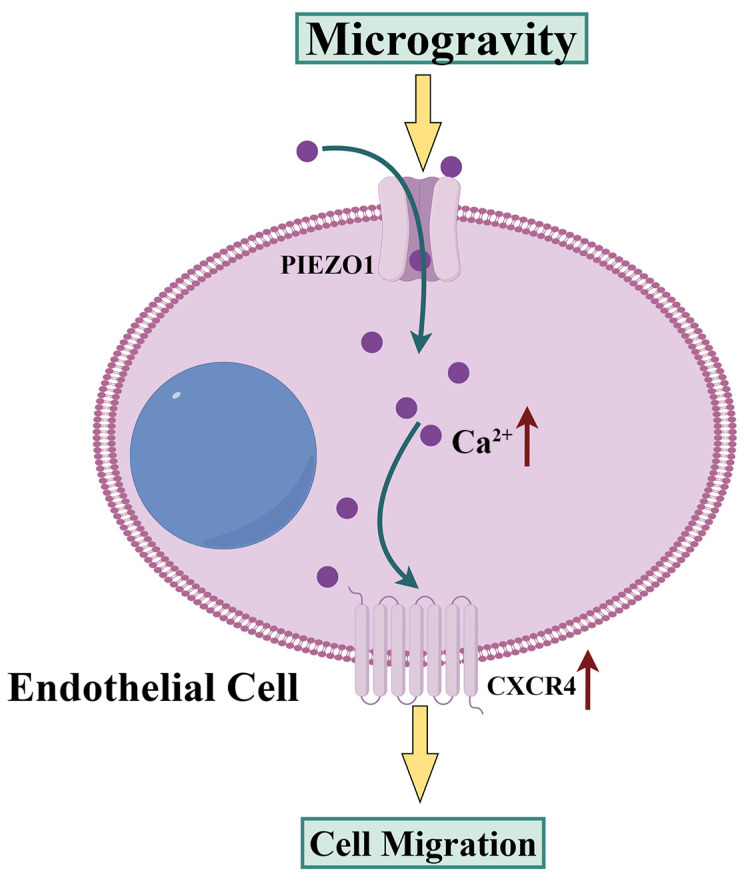
Schematic diagram of migration promotion via simulated microgravity in HUVECs. When HUVECs are exposed to simulated microgravity, PIEZO1 is upregulated. This upregulation of PIEZO1 facilitates the migration of HUVECs via enhancing CXCR4 expression through Ca^2+^ influx.

## Data Availability

The data that support the findings of this study are available from the corresponding author upon reasonable request.

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
