# Peer review of "PIEZO1 Promotes the Migration of Endothelial Cells via Enhancing CXCR4 Expression under Simulated Microgravity"

_ijms, 2024, doi:10.3390/ijms25137254_

Round 1
Reviewer 1 Report
Comments and Suggestions for Authors
The authors investigated the suppressive effect of the active form of vitamin D3 on the expression of PIEZO1, a mechanosensitive ion channel and its role for the migration in vascular endothelial cells. In my opinion, this paper has a challenging and interesting observation about the role of PIEZO1 for endothelial function such as migration, but the authors are overinterpreting the mechanisms of the induction of CXCR4 expression. I therefore recommend a major revision and re-review at this stage.
1. Although a proteomics analysis (as shown in figure 3) revealed that the expression of CXCR4 protein decreased in the PIEZO1-knocked down endothelial cells, there are no detailed analyzed data on the mechanisms of the regulation of CXCR4 expression beyond this finding. Therefore, stating that 'we showed that upregulation of PIEZO1 induced by simulated microgravity increased CXCR4 protein level through Ca2+ influx (in lines 228-229)' is an overstatement. A detailed analysis of the CXCR4 expression increase induced by microgravity is needed, using PIEZO1 knockdown, PIEZO1 agonists, and antagonists. Additionally, the involvement of increased calcium influx should also be analyzed.
2. It is preferable to discuss the mechanism by which microgravity increased PIEZO1 expression. Additionally, while FGF-2 and VEGF are well-known growth factors responsible for vascular endothelial proliferation and migration, it would be beneficial to also consider the significance of chemokine receptor involvement.
3. Regarding Figure 1B, the expression of GAPDH protein was significantly changed by microgravity in a time-dependent manner, making it difficult to determine if the Western blot analysis was performed correctly. The data should either be replaced or another internal control should be used. Additionally, it appears there are two bands for PIEZO1; it should be clarified which one corresponds to PIEZO1.
4. Regarding Figure 4A and 4C, it should be specified which protein band is being referred to.
5. Regarding Figure S1, the Figure S1 and the legend do not correspond correctly and should be revised.
6. Line 16: PIZEO1 → PIEZO1
Author Response
Response to Reviewer 1 Comments
|
||
1. Summary |
|
|
Thank you very much for taking the time to review this manuscript. Please find the detailed responses below and the in the re-submitted files. |
||
2. Major comments: |
||
Comments 1: Although a proteomics analysis (as shown in figure 3) revealed that the expression of CXCR4 protein decreased in the PIEZO1-knocked down endothelial cells, there are no detailed analyzed data on the mechanisms of the regulation of CXCR4 expression beyond this finding. Therefore, stating that 'we showed that upregulation of PIEZO1 induced by simulated microgravity increased CXCR4 protein level through Ca2+ influx (in lines 228-229) is an overstatement. A detailed analysis of the CXCR4 expression increase induced by microgravity is needed, using PIEZO1 knockdown, PIEZO1 agonists, and antagonists. Additionally, the involvement of increased calcium influx should also be analyzed. |
||
Response 1: Thank you for this precious and valuable comment. We have conducted a series of additional experiments to establish the mechanism that upregulation of PIEZO1 induced by simulated microgravity increased CXCR4 protein level through Ca2+ influx. Firstly, we have proved that the expression levels of CXCR4 protein increased significantly under simulated microgravity, but decreased after knocking down PIEZO1.Please see Figure 4A and Page 7, lines 182-185 in the revised manuscript. Secondly, we used Fluo-4AM staining to detect the intracellular Ca2+ of HUVECs. Exposure to simulated microgravity caused a significant increase in Ca2+ levels of HUVECs, which was dramatically attenuated by PIEZO1 knockdown treatment. Please see Figure 4B and Page 7, lines 187-192 in the revised manuscript. Then we used a permeable calcium chelator (BAPTA‐AM) to investigate the effects of Ca2+ levels on expressions of CXCR4 in HUVECs under simulated microgravity. Western Blot analysis showed BAPTA‐AM treatment significantly reduced the elevation of CXCR4 protein levels induced by simulated microgravity. Please see Figure 4C, Page 7, lines 192-196 in the revised manuscript. Taken together, these findings suggest that PIEZO1 significantly enhanced the expression of CXCR4 via Ca2+ influx under simulated microgravity. As for PIEZO1 antagonists, we used GsMTx4(2.5 µM) to treat HUVECs under simulated microgravity. Consistent with the results of PIEZO1 knockdown, GsMTx4 treatment reduced the elevated CXCR4 expression induced by simulated microgravity, as shown in the figure. This further substantiates the notion that PIEZO1 enhanced the expression of CXCR4 in endothelial cells under simulated microgravity. |
||
Comments 2: It is preferable to discuss the mechanism by which microgravity increased PIEZO1 expression. Additionally, while FGF-2 and VEGF are well-known growth factors responsible for vascular endothelial proliferation and migration, it would be beneficial to also consider the significance of chemokine receptor involvement. |
||
Response 2: We greatly appreciate your insightful suggestions and constructive feedback on our manuscript. We have expanded our discussion to include a more in-depth analysis of the possible mechanisms through which microgravity may increase PIEZO1 expression. Please see Page 11, lines 239-246 in the revised manuscript. Furthermore, we have tested FGF-2 and VEGF expression level. Western blot analysis showed that the expression of FGF-2 and VEGF in endothelial cells did not change following simulated microgravity, see following figure. This observation suggests that VEGFA and FGF-2 may not be involved in the migration of endothelial cells induced by simulated microgravity. Endothelial cell migration is a complex process involving the coordination of multiple molecules. Our previous research has shown that mitophagy inhibits endothelial migration by reducing the high expression of MMP1 under simulated microgravity conditions[1]. In this study, we have expanded upon the role of the PIEZO1/Ca2+/CXCR4 signaling axis in mediating the effects of simulated microgravity on endothelial cell migration. Our study is the first to demonstrate that simulated microgravity promotes endothelial cell migration through the activation of this signaling pathway. And This finding underscores the intricate regulatory mechanisms at play during endothelial cell migration.
Comments 3: Regarding Figure 1B, the expression of GAPDH protein was significantly changed by microgravity in a time-dependent manner, making it difficult to determine if the Western blot analysis was performed correctly. The data should either be replaced or another internal control should be used. Additionally, it appears there are two bands for PIEZO1; it should be clarified which one corresponds to PIEZO1. |
||
Response 3: Thank you for your meticulous review and for pointing out the concerns regarding Figure 1B in our manuscript. GAPDH is widely used as an internal control in studies involving simulated microgravity[1,2], and its expression doesn’t change under these conditions. The variation in GAPDH expression observed in Figure 1B is likely due to variations in sample loading and experimental methods that led to inconsistent normalization. We acknowledge that this could potentially mislead the interpretation of the data. Regarding the two bands observed for PIEZO1, we suspect that the additional band may be due to non-specific binding of the antibody. To avoid any ambiguity and to ensure the highest standards of data quality, we have conducted a thorough re-evaluation of the Western blot analysis, ensuring proper sample loading and standardized procedures to maintain consistency in the internal control. The results from these experiments have been used to replace Figure 1B in the revised manuscript. Comments 4: Regarding Figure 4A and 4C, it should be specified which protein band is being referred to. Response 4: Thank you for your meticulous review and for pointing out the oversight regarding the labeling of the bands in our Western blot images. We sincerely apologize for this lapse and appreciate the opportunity to correct it. The bands in question have now been clearly labeled in the revised manuscript. We have also taken this opportunity to thoroughly check all other figures and data for accuracy and completeness. Comments 5: Regarding Figure S1, the Figure S1 and the legend do not correspond correctly and should be revised. Response 5: I would like to express my sincere gratitude for your insightful review and for bringing to our attention the issues with the figure captions. We apologize for any confusion that our initial submission may have caused. Following your valuable suggestions, we have made the necessary corrections to the problematic figure captions in the revised supplementary (Please see Figure S1). We have ensured that all figure captions are now clear, accurate, and properly reflect the content of the images they accompany. Comments 6: Line 16: PIZEO1 → PIEZO1 Response 6: I am writing to convey my deepest apologies once again for the oversight in our previous submission. We appreciate your patience and guidance. As per your instructions, we have made the necessary revisions to our manuscript. We are committed to upholding the integrity and clarity of our research, and your feedback has been instrumental in helping us achieve that. We hope that these revisions will meet with your approval and that our manuscript will be considered for publication. Once again, thank you for your constructive criticism and for the opportunity to to address the issues you have identified. |
References:
- Li, C.; Pan, Y.; Tan, Y.; Wang, Y.; Sun, X. PINK1-Dependent Mitophagy Reduced Endothelial Hyperpermeability and Cell Migration Capacity Under Simulated Microgravity. Front. Cell. Dev. Biol. 2022, 10, 896014, doi:10.3389/fcell.2022.896014.
- Li, C.F.; Sun, J.X.; Gao, Y.; Shi, F.; Pan, Y.K.; Wang, Y.C.; Sun, X.Q. Clinorotation-induced autophagy via HDM2-p53-mTOR pathway enhances cell migration in vascular endothelial cells. Cell Death Dis. 2018, 9, 147, doi:10.1038/s41419-017-0185-2.

Reviewer 2 Report
Comments and Suggestions for Authors
The authors have set a series of experiments claiming that microgravity is responsible for piezo1 induction and several physiological responses of endothelial cells in microgravity. However, as the authors clearly stated, the cells have been in a rotator at a rate of 30 rotations/min, which results in a similar event as sheer stress. Since no controls from sheer stress alone were used and knowing that sheer stress is an activator of piezo1 expression doi 10.1016/j.actbio.2024.02.043, it is difficult to conceive that the data generated refers to microgravity. In addition, several other elements, such as the expression of CxCR4 by sheer stress, 10.1007/s12265-022-10235-y. which is also an unspecific marker. Thus, there are doubts about the role of these events in microgravity. In the experiments, it is unclear if Calcium ion concentrations are controlled. Moreover, the ko of piezo1 in the cells was not proven accordingly. The authors should analyze other cell marker expressions.
Comments on the Quality of English LanguageSeveral grammatical mistakes should be checked
Author Response
Response to Reviewer 2 Comments
1. Summary
Thank you very much for taking the time to review this manuscript. Please find the detailed
responses below and the corresponding revisions in the re-submitted files.
2. Point-by-point response to Comments and Suggestions for Authors
Comments 1: The authors have set a series of experiments claiming that microgravity is responsible for piezo1 induction and several physiological responses of endothelial cells in microgravity. However, as the authors clearly stated, the cells have been in a rotator at a rate of 30 rotations/min, which results in a similar event as sheer stress. Since no controls from sheer stress alone were used and knowing that sheer stress is an activator of piezo1 expression
doi 10.1016/j.actbio.2024.02.043IF: 9.7 Q1 , it is difficult to conceive that the data generated refers to microgravity. In addition, several other elements, such as the expression of CxCR4 by sheer stress, 10.1007/s12265-022-10235-yIF: 3.4 Q2 . which is also an unspecific marker. Thus, there are doubts about the role of these events in microgravity.
Response 1:
Thank you for your thorough review and constructive comments regarding our
manuscript. Your observation about the potential confusion between the effects of microgravity and shear stress on PIEZO1 induction is well noted.
To clarify, PIEZO1 is indeed a member of the mechanically sensitive ion channel family, capable of detecting extracellular mechanical pressures such as pressure, stretching, or shear stress, and converting these mechanical stimuli into electrical or chemical signals. In our experiments, we aimed to investigate the specific effects of simulated microgravity on PIEZO1 expression and endothelial cell physiology, distinct from the effects of shear stress. In response to your concerns, we would like to provide the following points: Distinction Between Simulated Microgravity and Shear Stress: We acknowledge that the rotatory motion at 30 RPM/min could be misconstrued as inducing shear stress. However, our
use of a 2D clinostat is intended to simulate microgravity conditions, not to apply shear stress. The clinostat operates by rotating the cells around a horizontal axis, which effectively randomizes the gravitational vector to approximately 10-3g, a value has been calculated using specific methods and is supported by data available on the DESC web site: http://www.desc.med.vu.nl.
The speed of the rotation is 30rpm/min(≈3rad/s) and the diameter of the rotation is 1cm, thus, the gravity simulated by the rotation is about 10-3g
according to the table. 1 ,2
Simulated Microgravity Setup: The 2D clinostat was specifically developed by the China
Astronaut Research and Training Center to simulate the effects of microgravity(A). Human
Umbilical Vein Endothelial Cells were incubated on coverslips within a chamber filled with
culture medium. Each chamber contains four coverslips, which are carefully inserted into
specially designed rackets to ensure that they remain stable and do not move during the
rotation process(B).
Avoiding Shear Stress Influence:
To further eliminate the influence of shear stress, we have taken precautions such as filling the culture flasks meticulously to avoid air bubbles and sealing them hermetically during rotation. These procedures were effective to ensure an
environment designed to simulated microgravity, not fluid dynamics-induced shear stress (C, D). Validation of the Model: The clinostat model we employed has been validated and widely used in research to simulate the effects of microgravity. The gravitational value of approximately 10-3g is a standard in the field and has been utilized in various studies, some of which have been published in reputable journals, such as follows: Gravity, a regulation factor in the differentiation of rat bone marrow mesenchymal stem cells.
IF11.0 Q1;
Alteration of calcium signalling in cardiomyocyte induced by simulated microgravity and
hypergravity. IF8.5 Q1;
Autophagy protects HUVECs against ER stress-mediated apoptosis under simulated microgravity.
IF7.2 Q1;
Clinorotation-induced autophagy via HDM2-p53-mTOR pathway enhances cell migration in vascular endothelial cells. IF9.0 Q1;
Effect of miR‑27b‑5p on apoptosis of human vascular endothelial cells induced by simulated
microgravity. IF7.2 Q1
We believe that these revisions will strengthen our manuscript and provide a clearer understanding
of the role of PIEZO1 in endothelial cell responses to simulated microgravity. We appreciate your guidance and look forward to your feedback on our revisions.
Comments 2: In the experiments, it is unclear if Calcium ion concentrations are controlled.
Response 2:
We appreciate your inquiry regarding the control of intracellular calcium ion
concentrations in our experimental groups.
In our experiments, we have taken meticulous steps to ensure the consistency of the starting conditions for all cell groups. Specifically, each cell group was cultured using
Dulbecco's Modified Eagle Medium (DMEM, Hyclone, USA) to ensure consistent initial Ca2+concentration(265mg/L).
Furthermore, we utilized Fluo-4AM staining to assess the intracellular calcium ion (Ca2+) content in the Con+si-NC (control group with non-targeting siRNA), MG+si-NC (simulated
microgravity group with non-targeting siRNA), and MG+si-PIEZO1 (simulated microgravity group with PIEZO1-targeting siRNA) groups. The results of this assay are presented in Figure 4B and detailed on Page 7, lines 189-192 of the manuscript. We believe that these data strengthen the evidence that PIEZO1 is involved in the mechanotransduction pathways of endothelial cells in response to simulated microgravity. The control of calcium ion concentrations is an integral part of our experimental design, ensuring the validity of our findings related to PIEZO1 function.
Comments 3: Moreover, the ko of piezo1 in the cells was not proven accordingly. The authors should analyze other cell marker expressions.
Response 3:
Thank you for your meticulous review and for pointing out the need for further
validation of PIEZO1 knockout (ko) in our study. We have indeed initially validated the knockdown efficiency of the siRNA targeting PIEZO1. We have screened multiple siRNAs for their efficacy in knocking down PIEZO1 expression. The validation was performed using Western blot analysis, and the results are
presented in Figure S1 on Page 4, lines 106-111 of the manuscript. After comparing the knockdown efficiency of three different siRNA sequences, we selected siRNA3 for its highest efficiency and used it for all subsequent experiments. We believe that these measures ensure the robustness of our experimental approach and the validity of our conclusions. We are
committed to providing a thorough and transparent account of our methods and findings. Thank you for your valuable feedback, which helps us to improve the quality and clarity of our research.
Comments 4: Several grammatical mistakes should be checked.
Response 4:
Thank you for your thorough review and for bringing the grammatical errors to our attention. We have taken your feedback seriously and have conducted a comprehensive review of our manuscript. Upon re-examination, we have identified and corrected all grammatical mistakes that were present in the original submission.
Thank you once again for your valuable input.

Reviewer 3 Report
Comments and Suggestions for Authors
The authors present an interesting study examining the effects of microgravity on endothelial function and health, with a specific focus on the role of PIEZO1 protein. Briefly, the authors utilise an in vitro, clinostat model of simulating microgravity on HUVEC cultures and measure baseline responses such as migration, calcium dynamics, and protein expression. These outcomes are then compared to experimental groups in which PIEZO1 protein has been knocked down by siRNA strategies, with the protein in question identified as playing a pivotal role in microgravity-driven changes in cell behaviour. In parallel, proteomics analyses identified CXCR4 as having an influence of PIEZO1 levels, and this is further validated through the use of a CXCR4 antagonist. Taken together, this was an interesting study that informs various areas of research.
In reviewing the manuscript however, I made a number of observations. The authors should address the following when preparing a suitable revision.
1. It almost looks like PIEZO1 exists as a doublet as evidenced in the difference in band size in figure 1 B. Was this taken into account/verified in any form of analyses?
2. The authors utilise a number of different time points throughout the study as exemplars of certain conditions within the 6-48 hour range. Is there a reason why certain timepoints were selected in certain instances? Were the assays such as migration and wound healing performed at other timepoints or only those presented?
3. The western blot in Figure 4 A and C need labelling.
4. In reviewing the original blot files some of the antibodies appear to detect several non-specific bands. In particular, CXCR4. Did the authors perform any kind of QC to ensure the band they selected for analysis was in fact that which they expected?
5. The signal presented in Figure 4B appears much stronger than what is represented in the bar chart. Can the authors clarify how they measured this signal and processed the results?
6. The scratches etched into the monolayers appear to be different sizes based on the lines drawn – did the authors take this into account in their analyses? How were the calculations performed to normalise these rate of closures to one another?
7. How was the working concentration of the CXCR4 inhibitor optimised? Is there an assay the authors performed to confirm that this compound was having the desired effect at the concentration used?
8. When the microgravity environment was being simulated was the culture vessel completed filled? How were the coverslips kept in place to ensure they weren’t subjected to movement as the clinostat operated?
9. What was the source of the primer sequences, and did the authors verify the primers were MIQE guidelines compliant?
10. More details on the image analyses of the microscopy images is warranted. Did the authors image several fields? Were they selected at random by the software? Etc.
11. Did the authors examine the effect of a nonspecific siRNA on PIEZO1 expression levels?
12. As the wound healing measurements were performed in normal gravity as is interpreted from the methods section, how can the authors be sure the effects of microgravity were sustained and reflected in the microgravity conditioned cells in the 12 hours after the assay began?
Author Response
Response to Reviewer 3 Comments
|
||
1. Summary |
|
|
Thank you very much for taking the time to review this manuscript. Please find the detailed responses below and the corresponding highlighted changes in the re-submitted files. |
||
2. Point-by-point response to Comments and Suggestions for Authors |
||
Comments 1: It almost looks like PIEZO1 exists as a doublet as evidenced in the difference in band size in figure 1 B. Was this taken into account/verified in any form of analyses? |
||
Response 1: Thank you for your careful review and for pointing out the potential issue with the band pattern observed in Figure 1B of our manuscript. We would like to confirm that PIEZO1 is not a doublet; rather, it has a homotrimeric structure, resembling a trident or three-leaf propeller. This trimeric assembly is fundamental to its function as a mechanosensitive ion channel[1].To address this, we have repeated the Western blot experiments with greater stringency in sample preparation and antibody selection. The new data have resolved the issue and confirm the expression of PIEZO1 as a single band corresponding to its expected molecular weight. We have replaced Figure 1B in the revised manuscript with the new, more accurate figure. Please see Figure 1B in the revised manuscript. Please see Figure 1 B and Page 3 in the revised manuscript. |
||
Comments 2: The authors utilize a number of different time points throughout the study as exemplars of certain conditions within the 6–48-hour range. Is there a reason why certain timepoints were selected in certain instances? Were the assays such as migration and wound healing performed at other timepoints or only those presented? |
||
Response 2: We greatly appreciate your thorough review and valuable comments on our manuscript. Our study employed a 2D clinostat to simulate the short-term microgravity effects on cells. Due to the limited oxygen in the rotating chamber, the maximum duration for continuous rotation is 72 h, and in general, 48h is used for rotation. While our previous studies have utilized a 48-hour simulated microgravity period[2,3], the current study selected 6-hour, 12-hour, 24-hour, and 48-hour time points to further elucidate the expression patterns of PIEZO1 following simulated microgravity conditions. For the Transwell and wound healing assays, we conducted our assessments at the 12-hour, without performing these assays at other time points. This decision was primarily based on the following considerations: 1.Literature Reference: Our approach aligns with relevant literature, which has adopted the 12-hour time point for such assessments, providing us with a reliable experimental framework[4,5]. 2.Experimental Conditions: During the experimental process, we used a low-serum medium to mitigate the impact of cell proliferation on the results. In contrast, by the 24-hour mark, we observed a considerable degree of cell detachment, which could potentially skew the results of the assays. The integrity and viability of the cells at the 12-hour time point were significantly better, making it the most appropriate choice for conducting our assays. Comments 3: The western blot in Figure 4 A and C need labelling. Response 3: Thank you for your meticulous review and for pointing out the oversight in our manuscript. We apologize for the missing labels in the Western blot images of Figure 4 A and C. The bands in question have now been clearly labeled in the re-submitted manuscript. We have also taken this opportunity to thoroughly check all other figures and data for accuracy and completeness. Comments 4:In reviewing the original blot files some of the antibodies appear to detect several non-specific bands. In particular, CXCR4. Did the authors perform any kind of QC to ensure the band they selected for analysis was in fact that which they expected? Response 4: Thank you for your careful review and insightful comments regarding our manuscript. We have taken note of the issue you raised concerning the non-specific bands detected by some antibodies, particularly CXCR4, in the original blot files. We acknowledge your concern and have conducted a thorough analysis to address this matter. Upon further examination, we believe that the observed non-specific binding could be attributed to the inherent properties of the antibodies used. However, we are confident that the band we selected for analysis in our manuscript is indeed the target band. To substantiate our claim, we referred to the antibody's product datasheet. According to the specifications provided, the CXCR4 band is expected to be around 43 kDa. Upon reviewing the original blot images, we observed that the predominant band aligns with this size, which is consistent with the expected molecular weight as per the datasheet.
Comments 5: The signal presented in Figure 4B appears much stronger than what is represented in the bar chart. Can the authors clarify how they measured this signal and processed the results? Response 5: Thank you for your attention to the details in our manuscript, particularly regarding the discrepancy between the signal intensity in Figure 4B and the bar chart representation. We appreciate the opportunity to clarify our methods and results. In our study, the experiments were conducted in three groups: Con+si-NC, MG+si-NC, and MG+si-PIEZO1. Each group consisted of three independent samples, and for each sample, we captured three different fields of view. To measure the signal, we utilized the Image J software to analyze the average fluorescence intensity of each group's samples. Following the quantification, we employed GraphPad Prism software to perform the statistical analysis on the collected data. The bar chart in the manuscript presents the final statistical results, which are the average fluorescence intensities of each group. The images shown in Figure 4B was selected to illustrate the most significant changes observed in one of the fields of view, which may account for the stronger signal appearance compared to the average values depicted in the bar chart. To avoid any ambiguity, we have replaced the figure with a more representative one. Please see Figure 4 B and Page 8 in the re-submitted manuscript We hope this explanation addresses your concerns and provides a clear understanding of our measurement and processing methods. Comments 6: The scratches etched into the monolayers appear to be different sizes based on the lines drawn – did the authors take this into account in their analyses? How were the calculations performed to normalize these rates of closures to one another? Response 6: Thank you for your close examination of our manuscript and for your specific inquiry about the scratch assay results presented in our study. We acknowledge that there may have been some inconsistencies in the scratch width due to potential operational errors during the process. The scratches were made using a 200 µl pipette tip, and variations could have occurred. To address this, we have calculated the scratch healing rate using the formula: scratch width at 0 hour - scratch width at 12 hours. This method allowed us to normalize the rates of closure despite the initial differences in scratch sizes. We believe this approach provides a fair assessment of the healing process over the 12-hour period. Furthermore, to ensure clarity and to avoid any ambiguity in our presentation, we have replaced the original images with ones that are more consistent and representative of our experimental findings. We have also included a detailed description of our scratch assay methodology and the normalization process in the revised manuscript. Comments 7: How was the working concentration of the CXCR4 inhibitor optimized? Is there an assay the authors performed to confirm that this compound was having the desired effect at the concentration used? Response 7: Thank you for your inquiry regarding the optimization of the working concentration of the CXCR4 inhibitor and the validation of its effect at the used concentration. AMD3100 is a selective CXCR4 antagonist that does not affect the expression levels of the CXCR4 receptor. Currently, there are no reports in the literature regarding the effective concentration of CXCR4 antagonists. Without objective indicators, the effectiveness of the antagonist can only be detected by observing downstream biological processes. In our study, we conducted a thorough literature review and found a broad range of concentrations reported for AMD3100, spanning from 5 µM to 700 µM[6,7]. To determine the appropriate concentration for our experiments, we selected a middle-ground concentration within this range. We chose 50 µM, which is a concentration that has been widely used in the literature for its effectiveness and lack of overt negative effects on cell morphology and growth[8,9]. At the 50 µM concentration, we observed a significant inhibition of endothelial cell migration under simulated microgravity conditions, without any apparent adverse effects on cell morphology or proliferation. This concentration has been employed in various studies, demonstrating its reliability and consistency in modulating CXCR4 activity. We believe that our selection of the 50 µM concentration for AMD3100 is well-justified and supported by both the literature and our experimental findings. We hope this response adequately addresses your concerns and provides a clear understanding of our approach to optimizing the working concentration of the CXCR4 inhibitor. Comments 8: When the microgravity environment was being simulated was the culture vessel completed filled? How were the coverslips kept in place to ensure they weren’t subjected to movement as the clinostat operated? Response 8: Thank you for your inquiry about the simulation of the microgravity environment in our study. To simulate the effects of microgravity, we employed a two-dimensional (2D) clinostat, which was specifically developed and provided by the China Astronaut Research and Training Center (A). Human Umbilical Vein Endothelial Cells were incubated on 2.55 × 2.15 cm coverslips within the chamber filled with culture medium. Each chamber contains four coverslips, which are carefully inserted into specially designed rackets to ensure that they remain stable and do not move during the rotation process(B). In order to avoid the influence of shear stress, all culture flasks were meticulously filled with medium to eliminate the presence of air bubbles and were hermetically sealed during rotation(C). The clinostat operates by rotating the cells around a horizontal axis at a speed of 30 rpm, which effectively randomizes the gravitational vector (D). This method is designed to mimic the microgravity conditions of low Earth orbit, which is approximately equivalent to 10-3 g of gravitational force. For the control group, the cells were cultured under the same conditions as the experimental group, with the exception of clinorotation, ensuring that any observed effects could be attributed to the simulated microgravity environment.
Comments 9: What was the source of the primer sequences, and did the authors verify the primers were MIQE guidelines compliant? Response 9: Thank you for your inquiry regarding the source of our primer sequences and their compliance with the MIQE guidelines. The primer sequences utilized in our study were sourced from the PrimerBank website, a reputable database that provides reliable and validated primers for various genes. For the genes of interest in our study, PIEZO1 and GAPDH, we ensured that the primers adhered to the following criteria in accordance with the MIQE guidelines: The GC content of both the PIEZO1 and GAPDH primers was maintained within the 40-60% range, which is essential for achieving optimal amplification efficiency and specificity. The melting temperatures (Tm) of the primers were confirmed to fall within the 50-65°C interval, ensuring that the primers would anneal effectively under standard PCR conditions. The PIEZO1 primers were designed to span exons 38 and 39, while the GAPDH primers spanned exons 8 and 9. This exon-spanning design helps to prevent the amplification of potential genomic DNA contamination. To further verify the specificity of our primers, we conducted sequence alignment using the NM sequences from the NCBI database. The results of this alignment confirmed that our primers were specific to their target genes, with no significant homology to other genomic regions. In summary, our primer design and selection process were meticulous and strictly followed the principles outlined in the MIQE guidelines[10]. We are confident that the primers used in our study are both specific and reliable, contributing to the validity and reproducibility of our research findings.
Comments 10: More details on the image analyses of the microscopy images are warranted. Did the authors image several fields? Were they selected at random by the software? Etc. Response 10: Thank you for your request for additional details regarding the image analysis of our microscopy images. We conducted three independent repetitions for each of the three experimental groups, with three samples per group. Each sample was randomly captured in at least three different fields of view for subsequent statistical analysis. In all cases, the image acquisition was performed in a blinded manner to minimize bias. The selection of fields was automated by the software to ensure randomness, and the analyses were conducted without prior knowledge of the experimental conditions. We hope this information provides the necessary details for your assessment and addresses your concerns about the image analysis process. Comments 11: Did the authors examine the effect of a nonspecific siRNA on PIEZO1 expression levels? Response 11: Thank you for your inquiry about the examination of the effect of a nonspecific siRNA on PIEZO1 expression levels. In the supplementary (Figure S1),we have included additional data that demonstrate our evaluation of three different siRNAs targeting PIEZO1. Through this assessment, we observed the extent of knockdown at both the mRNA and protein levels for each sinuate careful analysis, we selected the siRNA3 that exhibited the most pronounced reduction in PIEZO1 expression for our subsequent experiments. We believe that the inclusion of these data in the supplementary addresses the issue of siRNA specificity and strengthens the validity of our findings. We hope this information satisfies your query and provides the necessary evidence of the specificity of our siRNA approach.
Comments 12: As the wound healing measurements were performed in normal gravity as is interpreted from the methods section, how can the authors be sure the effects of microgravity were sustained and reflected in the microgravity conditioned cells in the 12 hours after the assay began? Response 12: We appreciate your concern about ensuring that the effects of microgravity are sustained and reflected in the microgravity-conditioned cells during the 12 hours following the initiation of the assay. We maintained strict experimental conditions for both the simulated microgravity group and the control group, with the only variable being the rotation to simulate microgravity. Our experimental design ensured that all other conditions, including the culture medium, cell seeding density, and incubation conditions, were identical for both groups. This rigorous control of variables allows us to confidently attribute any observed increases in endothelial cell migration to the simulated microgravity conditions. Furthermore, our assertion that the enhanced migration is due to simulated microgravity is supported by previous research conducted in our laboratory. Our previous studies have demonstrated similar effects of simulated microgravity on endothelial cells, which corroborates the findings observed in the current study[3,11]. We believe that the consistency in our experimental approach, combined with the supportive evidence from our previous research, provides a solid foundation for concluding that the effects observed in our study are indeed a result of the simulated microgravity. We hope this explanation clarifies how we ensured the sustained effects of microgravity in our study and addresses your concerns.
Thank you once again for your insightful comments and for giving us the opportunity to improve our work. |
References:
- Zhao, Q.; Zhou, H.; Chi, S.; Wang, Y.; Wang, J.; Geng, J.; Wu, K.; Liu, W.; Zhang, T.; Dong, M.Q. et al. Structure and mechanogating mechanism of the Piezo1 channel. Nature 2018, 554, 487-492, doi:10.1038/nature25743.
- Li, C.; Pan, Y.; Tan, Y.; Wang, Y.; Sun, X. PINK1-Dependent Mitophagy Reduced Endothelial Hyperpermeability and Cell Migration Capacity Under Simulated Microgravity. Front. Cell. Dev. Biol. 2022, 10, 896014, doi:10.3389/fcell.2022.896014.
- Li, C.F.; Sun, J.X.; Gao, Y.; Shi, F.; Pan, Y.K.; Wang, Y.C.; Sun, X.Q. Clinorotation-induced autophagy via HDM2-p53-mTOR pathway enhances cell migration in vascular endothelial cells. Cell Death Dis. 2018, 9, 147, doi:10.1038/s41419-017-0185-2.
- Xia, J.; Chu, C.; Li, W.; Chen, H.; Xie, W.; Cheng, R.; Hu, K.; Li, X. Mitochondrial Protein UCP1 Inhibits the Malignant Behaviors of Triple-negative Breast Cancer through Activation of Mitophagy and Pyroptosis. Int. J. Biol. Sci. 2022, 18, 2949-2961, doi:10.7150/ijbs.68438.
- He, X.; Lian, Z.; Yang, Y.; Wang, Z.; Fu, X.; Liu, Y.; Li, M.; Tian, J.; Yu, T.; Xin, H. Long Non-coding RNA PEBP1P2 Suppresses Proliferative VSMCs Phenotypic Switching and Proliferation in Atherosclerosis. Mol. Ther. Nucleic Acids 2020, 22, 84-98, doi:10.1016/j.omtn.2020.08.013.
- Xie, X.F.; Wu, N.Q.; Wu, J.F.; Zhang, G.L.; Guo, J.F.; Chen, X.L.; Du CW CXCR4 inhibitor, AMD3100, down-regulates PARP1 expression and Synergizes with olaparib causing severe DNA damage in BRCA-proficient triple-negative breast cancer. Cancer Lett. 2022, 551, 215944, doi:10.1016/j.canlet.2022.215944.
- Zhang, S.; Feng, F.; Dai, J.; Li, J.; Bu, X.; Xie, X. Recombinant High-Mobility Group Box 1 (rHMGB1) Promotes NRF2-Independent Mitochondrial Fusion through CXCR4/PSMB5-Mediated Drp1 Degradation in Endothelial Cells. Oxidative Med. Cell. Longev. 2021, 2021, 9993240, doi:10.1155/2021/9993240.
- Kim, J.; Lee, S.W.; Park, K. CXCR4 Regulates Temporal Differentiation via PRC1 Complex in Organogenesis of Epithelial Glands. Int. J. Mol. Sci. 2021, 22, doi:10.3390/ijms22020619.
- He, G.; Ma, M.; Yang, W.; Wang, H.; Zhang, Y.; Gao, M.Q. SDF-1 in Mammary Fibroblasts of Bovine with Mastitis Induces EMT and Inflammatory Response of Epithelial Cells. Int. J. Biol. Sci. 2017, 13, 604-614, doi:10.7150/ijbs.19591.
- Bustin, S.A.; Benes, V.; Garson, J.A.; Hellemans, J.; Huggett, J.; Kubista, M.; Mueller, R.; Nolan, T.; Pfaffl, M.W.; Shipley, G.L. et al. The MIQE guidelines: minimum information for publication of quantitative real-time PCR experiments. Clin. Chem. 2009, 55, 611-622, doi:10.1373/clinchem.2008.112797.
- Li, C.; Pan, Y.; Tan, Y.; Wang, Y.; Sun, X. PINK1-Dependent Mitophagy Reduced Endothelial Hyperpermeability and Cell Migration Capacity Under Simulated Microgravity. Front. Cell. Dev. Biol. 2022, 10, 896014, doi:10.3389/fcell.2022.896014.

Round 2
Reviewer 1 Report
Comments and Suggestions for Authors
The quality of the manuscript has greatly improved with more details and clarifications. I recommend the paper for publication in its present form.
Author Response
We would like to express our sincere gratitude for the insightful comments and
constructive suggestions provided. Your thorough review has significantly contributed to
enhancing the quality of our manuscript.
We have taken great care to address each point raised by you and have incorporated
additional details and clarifications as recommended. The revisions have led to a more
comprehensive and clearer presentation of our findings.
Based on the improvements made, we are confident that the manuscript is now
ready for publication in its current form. We believe that the paper's contribution to the
field is substantial and will be of interest to the readers of International Journal of Molecular
Sciences.
Thank you once again for your valuable input. We hope that you will find the
revised manuscript satisfactory for publication.
Reviewer 2 Report
Comments and Suggestions for Authors
The authors responded to the queries and made several changes in the manuscript. The manuscript is now suitable for publication
Comments on the Quality of English LanguageSeveral minor grammatical mistakes were encountered.
Author Response
We greatly appreciate the time and effort you invested in reviewing our manuscript. Your feedback has been instrumental in enhancing the quality and clarity of our work.
In response to your comments, we have thoroughly reviewed the entire manuscript for any grammatical inaccuracies. We have systematically identified and corrected all minor grammatical mistakes to ensure the manuscript meets the highest standards of language and presentation.
We believe that these revisions, along with the changes made in response to your earlier queries, have significantly improved the manuscript, making it now suitable for publication in International Journal of Molecular Sciences.
Reviewer 3 Report
Comments and Suggestions for Authors
The authors have suitably addressed the majority of my comments, and as such the manuscript is much improved/there is greater clarity around specific points. However, in terms of controlling the results, there remains some questions over the performance of some of the assays:
In reviewing the manuscript and the included blots for each still draw questions. For example, the blots performed for each examination of PIEZO1 are inconsistent with one another. The authors mention repeating the Western blot to address the original points raised, however, in reviewing those submitted with the article the bands obtained look varied between all included. For example, in some blots the band is almost like a smear and/or multiple bands are present, whereas in other the blots are much cleaner. I appreciate the authors are referring to the data sheet, but often these antibody checks have been performed using a particular cell line, and in such instances, every antibody should be validated for each respective model/system being employed regardless of the data sheet info. Have the authors ever included a positive/negative control to validate the bands they are analysing?
For MIQE guidelines validation, there are other aspects of primers that need validation. For example, clean single peak melt curves, product size validation, and efficiency falling between 90%-105% to name a few. Were any of these checks performed?
Until these questions are answered there are questions over some of the results.
Author Response
Response to Reviewer 3Comments
|
||
1. Summary |
|
|
Thank you very much for taking the time to review this manuscript. Please find the detailed responses below and the corresponding highlighted changes in the re-submitted files. |
||
2. Point-by-point response to Comments and Suggestions for Authors |
||
Comments 1: In reviewing the manuscript and the included blots for each still draw questions. For example, the blots performed for each examination of PIEZO1 are inconsistent with one another. The authors mention repeating the Western blot to address the original points raised, however, in reviewing those submitted with the article the bands obtained look varied between all included. For example, in some blots the band is almost like a smear and/or multiple bands are present, whereas in other the blots are much cleaner. I appreciate the authors are referring to the data sheet, but often these antibody checks have been performed using a particular cell line, and in such instances, every antibody should be validated for each respective model/system being employed regardless of the data sheet info. Have the authors ever included a positive/negative control to validate the bands they are analysing? Response 1: Thank you for your thorough review and constructive comments on our manuscript. We appreciate the concerns raised regarding the consistency of the Western blot results for PIEZO1. We assure you that all our results are reproducible and we have taken your feedback seriously to address the issues highlighted. We acknowledge that the variability observed in the PIEZO1 blots may be attributed to methodological variations during sample collection. To rectify this, we have recollected the samples and ensured a standardized protocol. Given that PIEZO1 is a membrane protein, we have adopted a milder denaturation approach to preserve its integrity during Western blot analysis. Regarding the presence of non-specific bands, we have included si-PIEZO1 samples as negative controls to validate the bands of interest. Upon re-examination, we found that these non-specific bands remained unaffected by the knockdown, suggesting that they are likely due to non-specific antibody interactions rather than representing PIEZO1 protein. Please see Figure S1 in the supplementary. We understand the importance of validating each antibody for the respective model or system used, regardless of the information provided on the data sheet. We have taken additional steps to ensure that all antibodies are rigorously tested and validated for the cell lines and models employed in our study. We hope that these revisions and additional controls will address the concerns raised and strengthen the validity of our findings. We are committed to maintaining the highest standards of scientific integrity and thank you once again for your insightful comments. Comments 2: For MIQE guidelines validation, there are other aspects of primers that need validation. For example, clean single peak melt curves, product size validation, and efficiency falling between 90%-105% to name a few. Were any of these checks performed? Response 2: Thank you for your meticulous review and for bringing to our attention the MIQE guidelines for primer validation. We appreciate the opportunity to clarify the measures we have taken to ensure the quality of our primers in our study. Before conducting our experiments, we meticulously ensured the quality of our primers, which is reflected in several key aspects: Firstly, we have examined the melt curves for all our primers and showed that they all exhibit clean, single peaks, indicating the specificity of our primers and the absence of primer-dimers in our reactions(A). Secondly, to validate the product size, we have utilized agarose gel electrophoresis to analyze the amplicons. The results confirm that the GAPDH product is 131 base pairs (bp) and the PIEZO1 product is 127 bp, which are in accordance with the predicted sizes (BC). Furthermore, we have calculated the amplification efficiency for both primers. The GAPDH primers show an efficiency of 91.5%, and the PIEZO1 primers an efficiency of 103%. Both values fall well within the recommended range of 90% to 110%(D). In summary, these validations confirm that our primers are suitable for use in our experiments, ensuring the accuracy and reproducibility of our qPCR data. We appreciate your guidance and are grateful for the opportunity to enhance the rigor of our study.
We hope that these revisions and additional controls will address the concerns raised and strengthen the validity of our findings. We are committed to maintaining the highest standards of scientific integrity and thank you once again for your insightful comments.
|
||
|
||
|

Round 3
Reviewer 3 Report
Comments and Suggestions for Authors
The authors have suitably addressed my comments.